# Highly Sensitive Fluorescence Sensor for Carrageenan from a Composite Methylcellulose/Polyacrylate Membrane

**DOI:** 10.3390/s20185043

**Published:** 2020-09-04

**Authors:** Riyadh Abdulmalek Hassan, Lee Yook Heng, Ling Ling Tan

**Affiliations:** 1School of Chemical Sciences and Food Technology, Faculty of Science and Technology, Universiti Kebangsaan Malaysia, 43600 UKM Bangi, Selangor Darul Ehsan, Malaysia; rydh1974@yahoo.com; 2Department of Chemistry, Faculty of Science, Ibb University, P.O. Box: 70270 Ibb, Yemen; 3Southeast Asia Disaster Prevention Research Initiative (SEADPRI-UKM), Institute for Environment and Development (LESTARI), Universiti Kebangsaan Malaysia, 43600 UKM Bangi, Selangor Darul Ehsan, Malaysia; lingling@ukm.edu.my

**Keywords:** fluorescence sensor, polymer blend, carrageenan, methylene blue

## Abstract

Carrageenans are linear sulphated polysaccharides that are commonly added into confectionery products but may exert a detrimental effect to human health. A new and simpler way of carrageenan determination based on an optical sensor utilizing a methylcellulose/poly(n-butyl acrylate) (Mc/PnBA) composite membrane with immobilized methylene blue (MB) was developed. The hydrophilic Mc polymer membrane was successfully modified with a more hydrophobic acrylic polymer. This was to produce an insoluble membrane at room temperature where MB reagent could be immobilized to build an optical sensor for carrageenan analysis. The fluorescence intensity of MB in the composite membrane was found to be proportional to the carrageenan concentrations in a linear manner (1.0–20.0 mg L^−1^, R^2^ = 0.992) and with a detection limit at 0.4 mg L^−1^. Recovery of spiked carrageenan into commercial fruit juice products showed percentage recoveries between 90% and 102%. The optical sensor has the advantages of improved sensitivity and better selectivity to carrageenan when compared to other types of hydrocolloids. Its sensitivity was comparable to most sophisticated techniques for carageenan analysis but better than other types of optical sensors. Thus, this sensor provides a simple, rapid, and sensitive means for carageenan analysis.

## 1. Introduction

The common use of carrageenan in food as an additive is safe at its intended level of load. However, the excess amount of carrageenan in food may cause harm to human health. Carrageenans have been extensively applied in the food and beverage industry, serving as thickener, stabilizer, emulsifier, and gelling agents. The safety of carrageenans for use in food was confirmed at the 57th meeting of the JECFA-Joint Food and Agriculture Organization of the United Nations/World Health Organization Expert Committee in Food Additives [1]. According to the JECFA, only degraded carrageenans are associated with adverse health effects, and should not be used as food additives. Recently, carrageenans have become an essential ingredient in pharmaceutical industries to reduce the amount of polymorphic transformation in tableting [2], and to produce controlled release matrix tablets [3] to stimulate interactions with drugs for modified release systems [4].

The purity and composition of carrageenan in commercial food samples may vary significantly. Therefore, there is a need to authenticate their composition and concentrative properties in various edible products. Various approaches are available for assessment of carrageenan content in foods, especially confectionery goods. These include colorimetric staining for total carrageenan determination, light microscopy, immunological detection, calorimetry, fluorimetry, electrophoresis, nuclear magnetic resonance (NMR), and chromatographic methods. Chromatographic methods are more often used, e.g., gas−liquid chromatography (GLC), high-performance liquid chromatography (HPLC) spectroscopy, gel permeation coupled with inductively coupled plasma-atomic emission (GPC/ICP-AES) [5], and high-performance anion-exchange chromatography coupled with chemical or enzymatic depolymerization procedures [6,7].

Spectrophotometric methods for the detection of carrageenan in the liquid state are based on cationic dyes, which are hydrophilic substances for reaction with anionic carrageenan. The requirement of digestion and addition of chemicals are the major problems in this method. The linear detection range of the spectrophotometric method for carrageenan is normally narrow and in the low concentration range. ELISA and dot-blot assay were used to assay kappa- (κ) carrageenan and were proved to be sensitive and specific with no cross-reaction with other thickeners. The limit of κ-carrageenan detected was in the range of 0.001–0.010% *w*/*w* in foodstuffs [8]. Another report using a sandwich ELISA for the determination of *κ*-carrageenan could yield a linear concentration range of 16−256 ng mL^−1^ [9], while the electrophoretic method using laser-excited indirect fluorescence (LIF) means with Schiff’s reagent and toluidine blue staining for iota- (ι) and *κ*-carrageenan could give a mass limit of detection of 95 ng. This indirect LIF method using fluorescein for carrageenan also provided a low detection limit in the picogram range in some cases [10].

Many of the methods mentioned above have demonstrated a satisfactory lower detection limit for the analysis of carrageenan in foods. However, the disadvantages of using these conventional analytical instruments are the need for complicated sample pre-treatment and elaborate operating procedures, hence the need for skillful operators. Such techniques are also expensive and time-consuming due to the need for complicated isolation, derivatization, and purification before the analysis can be performed. Furthermore, some of these reported methods are not specific, and supplementary methods must be used to determine the type of carrageenan correctly.

In this work, a new optical sensor based on the change in fluorescence emission intensity was developed to determine the concentration of carrageenan at a low limit of detection that is comparable to many sophisticated instrumental techniques. The optical change was induced through strong chemical interactions rather than simple physical adsorption, resulting in fluorometric changes (i.e., chemoresponsive). A methylcellulose/poly(n-butyl acrylate) (Mc/PnBA) composite membrane was employed for the first time in the development of such an optical sensor. The polyacrylate membrane was blended with the Mc polymer to increase the overall hydrophobicity of the membrane to improve its usefulness for carrageenan sensing. Methylene blue (MB) was used as a carrageenan sensing fluorogenic dye and immobilized in the composite membrane. Carrageenan was detected by the electrostatic interaction between the cationic site of immobilized MB, i.e., alkylamino (=NR_2_^+^) functional group and anionic site of carrageenan, i.e., the negatively charged sulphate (−SO_4_^−^) functional group. As a result of such interaction, the blue-colored membrane turned purple due to the formation of a metachromatic complex. This will lead to a change in the fluorescence intensity, as well as the color of the sensor membrane. The MB could be immobilized in the Mc/PnBA membrane likely via electrostatic interactions, hydrogen bonding, and van der Waals forces [11] between the membrane structure and MB molecules. Figure 1 illustrates the chemical reaction between the immobilized MB and carrageenan, and the chemical structure of the Mc/PnBA composite membrane.

## 2. Experimental

### 2.1. Materials and Apparatus

Tris-HCl buffer was prepared by using tris(hydroxymethyl) aminomethane (THAM, Acros Organics, Belgium, USA) in Milli-Q water, and hydrochloric acid 37% (HCl, Riedel-de Haen, Seelze, Germany) was added to adjust the Tris-HCl buffer to the required pH. A stock solution of methylene blue (MB, R & M Chemicals, Selangor, Malaysia) at 1 mM was prepared in Milli-Q water and stored at 4 °C in the dark. Dilution of MB solution was done using dimethylformamide (DMF). The analytical grade of iota- (ι, Sigma, Laramie, USA), lambda- (λ, Sigma), and kappa- (κ, Fluka, Buchs, Switzerland) carrageenans including calcium alginate, starch, and gum Arabic from Fluka was prepared by dissolving 50 mg of each carrageenan sample in 50 mL of Milli-Q water in a water bath at 50 °C. Other chemicals were of analytical reagent grade and used without purification.

The fluorescence intensity of MB was observed by using an optical Perkin Elmer Fluorimeter 4002 at an emission wavelength of 675 nm. The preparation of acrylic polymer was done by photopolymerization reaction by utilizing an UV exposure unit (RS Ltd., Cambridge, UK), which comprised four light tubes (15 W UV) transmitting ultraviolet (UV) radiation at an absorption wavelength of 350 nm under continuous nitrogen gas purging. Chemical characterization of the Mc/PnBA polymer blend was performed on a Perkin Elmer Spectrum GX FTIR microscope using the KBr disc method. A scanning electron microscope (SEM, LEO 1450VP) was used to characterize the morphology of the composite membrane. The successful immobilization of MB on the Mc/PnBA composite membrane was confirmed by SEM-EDS (scanning electron microscope with an energy dispersive X-ray spectrometer) that detected the presence of MB dye in the film sample.

### 2.2. Preparation of Methylcellulose/Polyacrylate Optical Sensor Membrane

Polymers used in this study were methylcellulose (Mc) and poly(n-butyl acrylate) (PnBA). Polymer solutions of 2% *w*/*v* PnBA and Mc in THF were prepared and used separately in stoppered conical flasks. Similarly, different polymer blend compositions were prepared by mixing appropriate quantities of stock solutions of Mc and PnBA, and their blend compositions at 100/0, 80/20, 70/30, 50/50, 30/70, and 20/80% (*v*/*v*) were prepared in THF under vigorous stirring for one day. The MB immobilization process was carried out by mixing 100 µL of the MB in DMF to 2 mL of polymer blend solution and stirred overnight. Finally, 20 µL of MB immobilized polymer blend was dropped onto a flat piece of a glass surface, and the sensor was left overnight at room temperature (25 °C). The dried sensor was then washed thoroughly with Tris-HCl buffer (pH 7) and kept at 25 °C in a dark and dry place before further usage.

### 2.3. Optimization of the Carrageenan Fluorescence Sensor Response

The concentration−effect study on the immobilized MB was carried out by immobilizing MB dye onto the composite membrane at different concentrations from 0.01 to 0.15 mM, and the fluorescence intensity of the sensor membrane was measured at 675 nm. For the effect of pH on the analytical sensing performance of the carrageenan fluorescence sensor, three series of λ-carrageenan solutions in the concentration range of 5.0–20.0 mg L^−1^ were prepared in 20 mM Tris-HCl buffer at pH 4, pH 7, and pH 9, and examined with the optode. The buffer concentration effect was conducted by preparing 10.0 mg L^−1^ λ-carrageenan in Tris-HCl buffer (pH 7) in the concentration range of 10−100 mM. The linear response range of the optical sensor for κ-, ι-, and λ-carrageenans was determined using 0.001−25.000 mg L^−1^ carrageenan solution in 20 mM Tris-HCl buffer (pH 7). Anionic polysaccharides such as starch, calcium alginate, and gum Arabic in the concentration range of 0.001–25.000 mg L^−1^ were also prepared to determine the selectivity of the optical fluorescence sensor. Sensor performance toward carrageenan determination was evaluated by spike-and-recovery tests using spiked standards of λ-carrageenan at 5.0, 10.0, and 15.0 mg L^−1^. The sample used was commercial beverages such as pineapple, apple, and orange juices. The fluorescence response of the sensor was measured at 675 nm. All the sensor characterization experiments were performed in triplicate.

## 3. Results and Discussion

### 3.1. Morphology of the Methylcellulose/Poly(n-butyl acrylate) Composite Membrane

The uniformity of the dispersion of solution-cast films of different Mc/PnBA blend compositions was examined through SEM, and examples of the micrographs are shown in Figure 2. At a low content of PnBA in the composite membrane, Mc/PnBA was dispersed on the film completely and showed a single phase. Conversely, when the PnBA content was increased, the films showed a rough surface, which can be observed in the SEM image [12]. For instance, at 80/20% (*v*/*v*) Mc/PnBA blend composition, the film showed a good smooth surface. Due to the hydrophilic character of Mc, which tends to dissolve in water, a high amount of Mc composition in the polymer blend would cause the formation of pores on the surface. By contrast, large agglomerates were observed for the blended sample containing higher PnBA content, e.g., 20/80% (*v*/*v*) Mc/PnBA, which showed incompatibility between Mc and PnBA (Figure 2f). The composite membrane at 70/30% (*v*/*v*) Mc/PnBA was chosen as the best film composite because it possessed good adhesion properties and insolubility in water [13,14]. The mixture of these two polymers gave a smooth and coherent surface with a good dispersion on the glass slide. The 70/30% (*v*/*v*) Mc/PnBA polymer blend had improved the membrane properties in terms of membrane water insolubility. Thus, there was no disintegration of the membrane in water. Besides, a slightly higher amount of Mc had allowed more interaction of the resulting membrane with MB dye and improved the immobilization process. This can subsequently be beneficial for carrageenan binding.

The immobilization of MB in the polymer composite of 70/30% (*v*/*v*) Mc/PnBA could be performed via hydroxyl (-OH) functional groups of the Mc polymer. This polymer composite contains a large number of -OH groups from Mc, and these functional groups could bind to MB by physicochemical interactions, which are mainly via ion exchange or complex formation between dyes and the functional groups of Mc [15]. This is due to the fact that cellulose and its derivatives are relatively inert except their hydroxyl groups, which are responsible for the majority of the reactions with organic and inorganic reagents, and this leads to inter- and intra-molecular hydrogen bonding [16].

The immobilization of MB on the blended composite is to prevent aggregation and self-quenching of the MB molecule, which decreases the fluorescence intensity. The successful immobilization of MB on the Mc/PnBA composite film was confirmed by SEM-EDX that detected the presence of MB in the film sample. MB and carrageenan containing sulfur elements were mapped by scanning electron microscopy (SEM) with the SEM-EDS spectrum for polymer composite-MB (Figure 3a). MB concentration was chosen for the best intensity to be suitable as a carrageenan sensor. The SEM-EDS spectrum showed the presence of sulfur element after addition of 100 mg L^−1^ of λ carrageenan (Figure 3b). The SEM-EDS spectrum exhibited an increment in the sulfur element of the film after addition of the λ-carrageenan sample.

### 3.2. FTIR Analysis of Mc/PnBA Composite Membrane

The superimposed FTIR spectra of PnBA, Mc, and Mc/PnBA polymers are shown in Figure 4. The presence of characteristic bands in the final polymer blend (Figure 4c), which were initially present in the respective starting materials, i.e., PnBA (Figure 4a) and Mc (Figure 4b), confirmed the formation of the Mc/PnBA polymeric composite. In Figure 4c, the absorption band at 1160 cm^−1^ is associated with the anti-symmetric stretching vibration of the C-O-C bridge of Mc. A broad O-H absorption band at 3426 cm^−1^ and a sharp C=O absorption band of aldehyde appeared at 1645.3 cm^−1^, also implying the presence of Mc in the polymer blend. The C=O ester functional group from the acrylic compound corresponds to the absorption band at 1730.1 cm^−1^ [17].

### 3.3. Optimization of Immobilized MB Concentration

The emission peaks of the immobilized MB in the composite membrane at different concentrations from 0.01 to 0.15 mM were measured with the fluorimeter to obtain the maximum fluorescence signal at 675 nm. The fluorescence intensity of the immobilized MB was found to increase from 0.01 to 0.05 mM as a larger amount of MB fluorophore dye was bound to the composite membrane, thus affording a higher fluorescence emission (Figure 5). It is important to note that MB binding is immediate and requires no incubation step, whereby the MB compound was physically attached on the self-adhesive membrane, thereby simplifying the reagent immobilization method [18]. As the concentration of immobilized MB increased beyond 0.05 mM, the fluorescence signal gradually declined from 0.05 to 0.15 mM. This might be due to the self-quenching effect between MB monomers and dimers at high MB concentration loading [19]. Therefore, 0.05 mM immobilized MB was used in the next carrageenan optical sensor optimization experiments.

### 3.4. Effect of Buffer pH on the Carrageenan Optosensor Response

The formation of a metachromatic complex is dependent on the pH, ionized acidic groups, and ionic strength of the reaction medium [20]. The variation in pH has an ionic strength impact on the adsorption interaction of MB cationic dye [21,22,23] and metachromatic complex formation via electrostatic interaction due to the changes in the ionic state of the polyanions’ functional groups. Besides, changes in pH can also influence the substrate binding with MB and ultimately affects the immobilization reaction of MB [21,24]. The colorimetric determination of carrageenan and other anionic hydrocolloids with MB dye in solution has been carried out by Soedjak [24]. The anionic sites of the hydrocolloids appeared to be primarily responsible for the dye-binding to form the purple metachromatic complexes (maximum absorption at 559 nm). The interaction is assumed electrostatic with a 1:1 stoichiometric ratio between the anionic sites and the bound dye molecules.

The optical sensor based on the MB-immobilized Mc/Pn-BA membrane has been used to determine λ-carrageenan in the concentration range of 5.0–20.0 mg L^−1^ in 20 mM buffer at various pHs. In acidic pH, i.e., pH 4, the immobilized MB and λ-carrageenan underwent protonation at =NR_2_^+^ and sulfate functional groups, respectively. As the λ-carrageenan became partially uncharged at low pH [25], it slowed down the electrostatic interaction between λ-carrageenan and MB. By contrast, under the alkaline conditions at pH 9, both immobilized MB and λ-carrageenan were deprotonated, and the reaction between optode and analyte was not chemically favored. The most suitable pH condition for the immobilized MB indicator dye to form the metachromatic complex with a carrageenan polyanion was at pH 7, as an obvious color change from blue to purple hue was observed at near-neutral conditions [26,27,28]. As Table 1 indicates, the highest sensitivity of the optical sensor toward carrageenan detection within the linear λ-carrageenan concentration range of 5.0–20.0 mg L^−1^ occurred at pH 7 with a correlation coefficient (R^2^) obtained at 0.9923. Thus, all subsequent optimization studies were carried out at pH 7 using 20 mM Tris-HCl buffer.

### 3.5. Effect of Buffer Concentration

The buffer capacity is an important variable that must be optimized to ensure adequate ionic strength and charge balance of the reaction buffer to permit a repeatable and maximum fluorescence response to be obtainable by the proposed optical sensor. In this study, the fluorescence intensity of the sensor in the detection of 10.0 mg L^−1^ λ-carrageenan using varying concentrations of Tris-HCl buffer (pH 7) was determined. Figure 6 represents the buffer concentration effect on the fluorescence response trend of the optosensor. The highest fluorescence response was obtained at 20 mM Tris-HCl buffer, after which the fluorescence intensity dramatically dropped between 20 and 100 mM Tris-HCl buffer (pH 7). The high ionic strength of the buffer appeared to distort the ionic atmospheres of both ionic reactant and analyte. It resulted in the conformational change in the polyanion (the charged polymer molecule) [29,30], and this has significantly changed the selectivity of the selective chemical layer toward its target analyte.

### 3.6. Optical Sensor toward Carrageenan Detection

As the concentrations of the carrageenan increased, greater electrostatic interaction between immobilized MB and carrageenan polyanions occurred. A light purple metachromatic complex was formed. The complexation between immobilized MB and carrageenan was formed between the cationic functional group of MB, i.e., =NR_2_^+^, and the negatively charged sulphate functional group of carrageenan. The proposed optode can detect carrageenan by the fluorescence intensity change of the MB immobilized in the composite membrane. As in Figure 7a, the fluorescence intensity of the optical sensor decreased upon reaction with the increase in carrageenan concentrations from 0.001 and 1.000 mg L^−1^. This was attributed to less electrostatic interaction between immobilized cationic MB and anionic carrageenan where most of the immobilized MB molecules remained uncomplexed, thus giving a higher fluorescence emission response. The fluorescence sensor was found to respond linearly to the increasing carrageenan concentration between 1.0 and 20.0 mg L^−1^ (Figure 7a) with a good linear R^2^ value of close to + 1 (Table 2) and reproducibility (relative standard deviation, RSD) of <5% (n = 3). The limit of detection (LOD) of the sensor was 0.4 mg L^−1^. It was evaluated from three times the standard deviation of the mean intensity of the blanks and the calibration curve of carrageenan. The optical carrageenan sensor reported here is a single-use sensor strip of low cost for rapid analysis of carrageenan. Due to the relatively low cost needed, it is not necessary to reuse the sensor, as the regeneration of the sensor for further use may take more time than the analysis.

The R^2^ is the linear correlation coefficient between the fluorescent intensity of the sensor and the concentrations of the carrageenan measured (Table 1 and Table 2). A value of R^2^ = 0.99 means that 99% of the two sets of data are correlated in a certain range of concentrations. Thus, the concentration is strongly correlated with the fluorescent intensity in a linear manner. The R^2^ value can be calculated by a correlation statistical program via the data processing software Excel Spreadsheet from a plot of fluorescent intensity against the carrageenan concentrations.

The sensitivity of the sensor as mentioned in Table 1 and Table 2 is defined by the response of the sensor, i.e., the fluorescent intensity per unit change in concentration. Thus, this is a value of the slope from the linear plot of intensity vs. carrageenan concentration over a certain concentration range. Thus, the larger the magnitude of the slope (or sensitivity), the more sensitive the sensor response (fluorescence) toward changes in concentration.

Besides, the optical sensor showed slightly different response factors for the detection of the respective κ-, ι-, and λ-carrageenans according to the molar sulfate content of carrageenans at pH 7 with 20 mM Tris-HCl buffer. No change in response of the sensor was observed toward other anionic polysaccharides such as starch, calcium alginate, and gum Arabic even at different concentrations. This indicates that little reaction occurred between the immobilized MB and those polysaccharides. Thus, the optical sensor was selective toward the detection of carrageenan (Figure 7b). The selectivity behavior of the sensor is similar to a study on carrageenan and other anionic hydrocolloids analysis using MB by the UV-Vis spectrophotometric method [24].

### 3.7 Recovery Studies of the Fluorescence Carrageenan Optical Sensor

A recovery study was performed to validate the performance of the proposed carrageenan optical sensor by using real samples spiked with a known amount of the analyte. A fixed amount of ɩ-carrageenan was added into pineapple, apple, and orange juices. The concentration used was between the linear range of the sensor calibration curve. Based on the data tabulated in Table 3, the sensor showed percentage recoveries of carrageenan ranged between 90% and 102%. This suggests that common additives that are found in the commercial juice products have little or no interference effect on the analytical sensing performance of the carrageenan fluorescence sensor.

To our best knowledge, so far, no one has reported on the carrageenan detection using the fluorometry approach. Thus, it is not possible to compare to a similar fluorescent sensor reported. However, it is still fair to compare to other optical sensors based on reflectance or UV-Vis spectrophotometry (Table 4) because they also employed immobilized MB as a sensing material [31]. Most of the current methods for carrageenan analysis involve instrumental procedures, e.g., electrophoresis, chromatography methods, and spectrophotometric methods. Therefore, equipment and chemical consumables contributed to the high cost for carrageenan analysis. Our estimation of the sensor cost is about 10 times lower than that for an average cost of analysis via instrumentation techniques. Clearly, the fluorescence sensor based on the MB-immobilized Mc/PnBA composite membrane demonstrated an LOD performance toward carrageenan with a significantly lower detection limit by a factor of approximately 2000. Hence, the fluorescence-based carrageenan sensor has achieved a low LOD comparable to that of most established methods, such as immunoassay and electrophoresis, for the analysis of carrageenan in foodstuffs.

## 4. Conclusions

The hydrophilic methylcellulose polymer was successfully modified with a more hydrophobic poly(n-butyl acrylate) to yield an insoluble membrane. This membrane functioned as a fluorescence sensor for assay of carrageenan where methylene blue could be immobilized in the Mc/PnBA polymer blend. The fluorescence carrageenan sensor demonstrated good selectivity and sensitivity toward carrageenan, and has a high potential for simple, rapid, and selective analysis of carrageenan in food samples. Therefore, extensive sample pre-treatment may not be required as compared to most of the currently used techniques. The sensitivity of the optical sensor is comparable to many standard methods for carrageenan analysis of various food items.

## Figures and Tables

**Figure 1 sensors-20-05043-f001:**
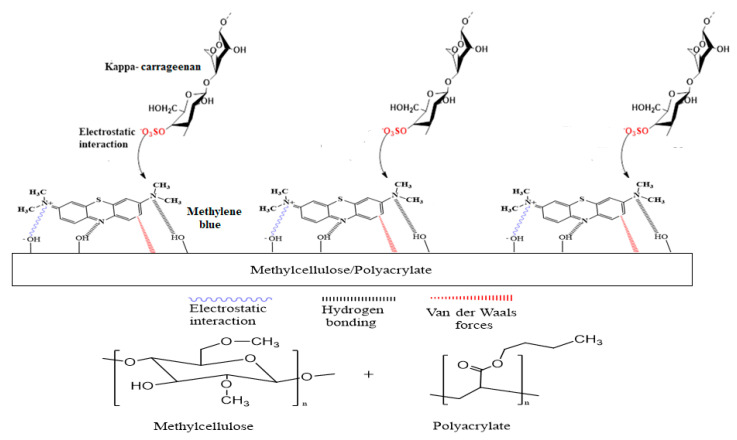
Schematic representation of a chemical reaction between a cationic functional group of methylene blue (MB) (=NR_2_^+^) and anionic functional group of carrageenan (SO_4_^−^), and the chemical structures of methylcellulose and poly(n-butyl acrylate). Blue-colored membrane changed to a light purple hue after the addition of a carrageenan solution.

**Figure 2 sensors-20-05043-f002:**
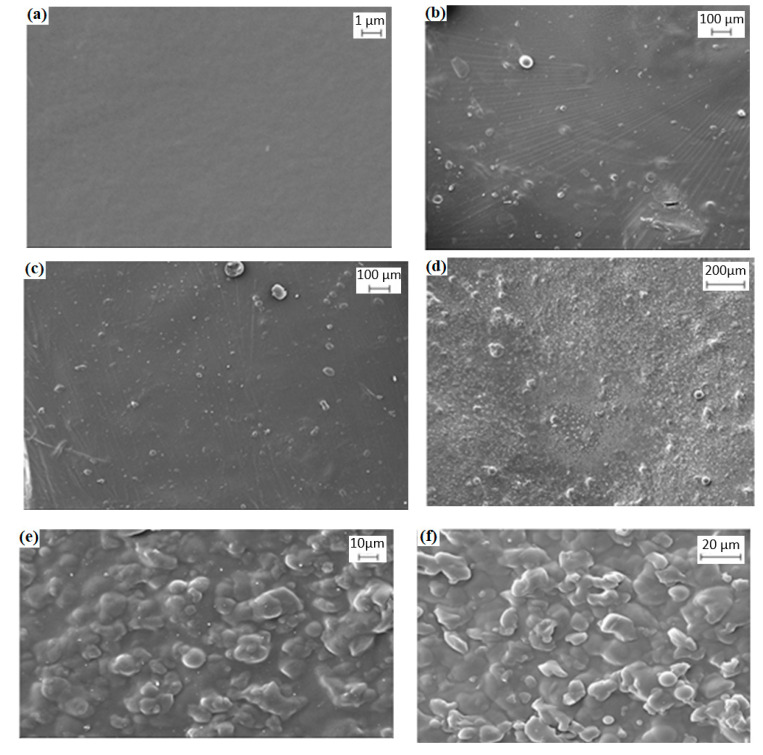
SEM micrographs of methylcellulose/poly(n-butyl acrylate) (Mc/PnBA) polymer blend with various Mc/PnBA compositions: (**a**) 100/0, (**b**) 80/20, (**c**) 70/30, (**d**) 50/50, (**e**) 30/70, and (**f**) 20/80.

**Figure 3 sensors-20-05043-f003:**
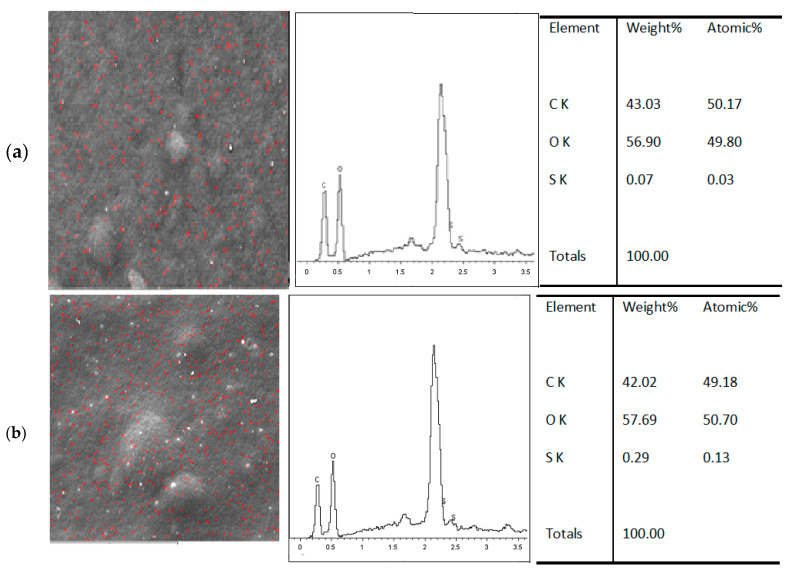
(**a**) SEM-EDS spectrum analysis result of S, O elements in the MB immobilized Mc/PnBA polymer composite, and (**b**) the presence of S element on the MB-Mc/PnBA membrane before and after addition of 100 mg L^−1^ λ -carrageenan.

**Figure 4 sensors-20-05043-f004:**
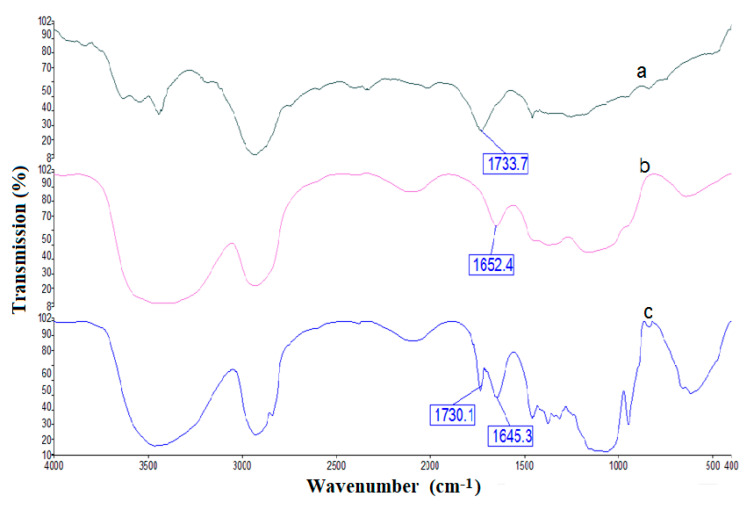
FTIR spectra for (a) PnBA, (b) Mc, and (c) Mc/PnBA composite membranes.

**Figure 5 sensors-20-05043-f005:**
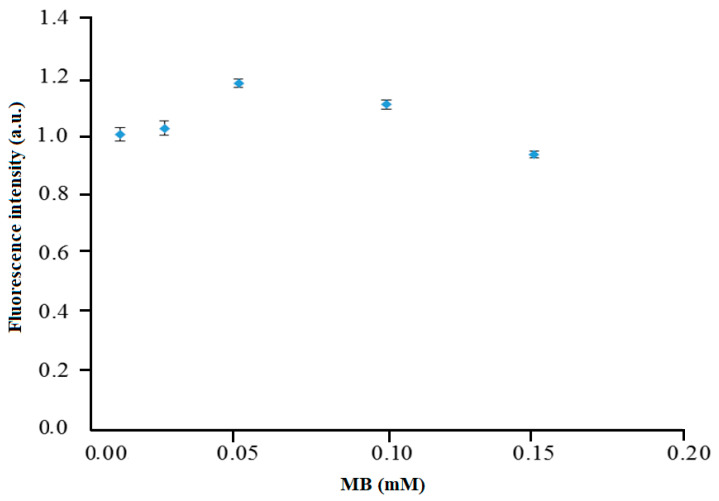
Effect of immobilized MB concentration on the fluorescence intensity of the Mc/PnBA composite membrane. Error bars represent data from 3 identical sensor runs and triplicate measurements of each.

**Figure 6 sensors-20-05043-f006:**
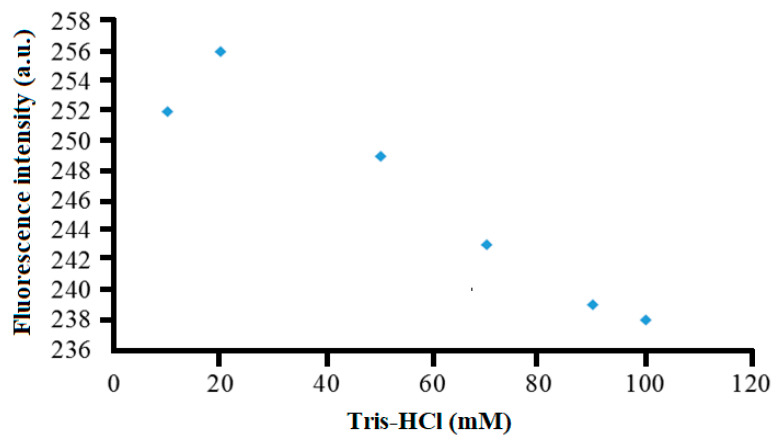
The effect of Tris-HCl buffer concentration (pH 7) on the fluorescence response of the optical sensor toward the detection of 10 mg L^−1^ λ-carrageenan at 675 nm.

**Figure 7 sensors-20-05043-f007:**
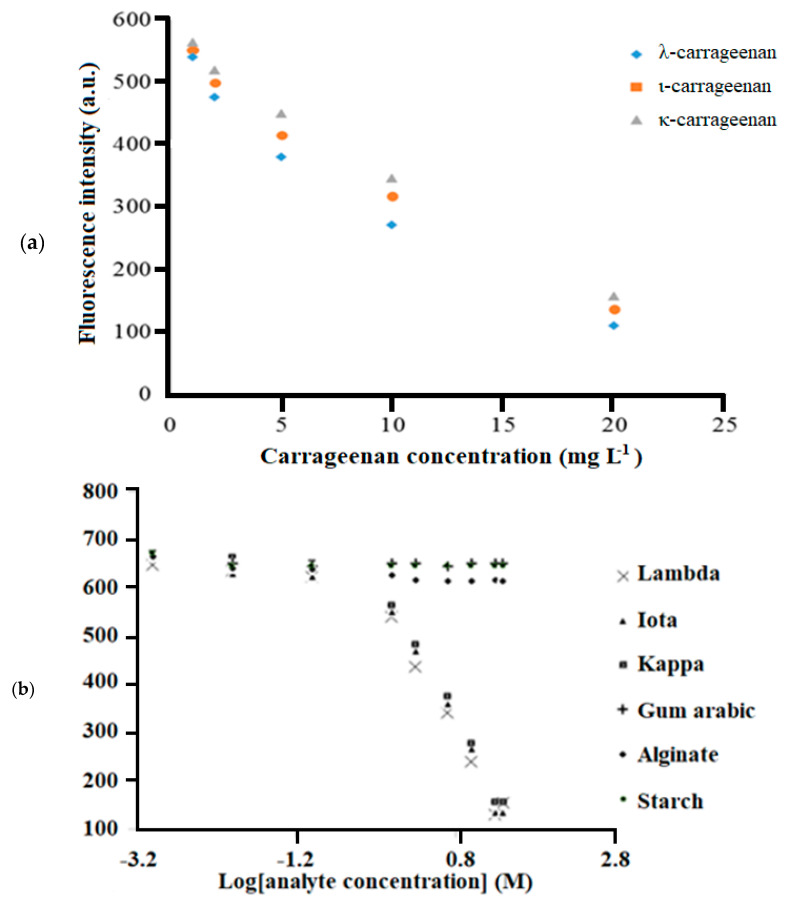
(**a**) The linear responses of the optical sensor for κ-, ι-, and λ-carrageenans determined at an emission wavelength of 675 nm. (**b**) The fluorescence response curve of the Mc/PnBA membrane-based optical sensor toward the detection of κ-, ι-, and λ-carrageenans, as well as anionic polysaccharides such as starch, calcium alginate, and gum Arabic.

**Table 1 sensors-20-05043-t001:** The R^2^ and sensitivity values of the fluorescence optosensor in the determination of 5.0−20.0 mg L^−1^ λ-carrageenan at different pH conditions.

pH Value	R^2^	Sensitivity	λ-Carrageenan Linear Concentration Range (mg L^−1^)
4	0.9815	−407	5.0–20.0
7	0.9923	−564	5.0–20.0
9	0.9419	−434	5.0–20.0

**Table 2 sensors-20-05043-t002:** The performance of the optical sensor for the analyses of various types of carrageenan.

Types of Carrageenan	R^2^	Sensitivity	Linear Range (mg L^−1^)
κ	0.98	−312	1.0–20.0
ι	0.99	−304	1.0–20.0
λ	0.99	−300	1.0–20.0

**Table 3 sensors-20-05043-t003:** Recovery of carrageenan from spiked food samples using fluorescence carrageenan optosensor (*n* = 3).

Spiked ɩ-Carrageenan Concentration(mg L^−1^)	Found(mg L^−1^) Pineapple Juice	*R (%)	Found (mg L^−1^) Apple Juice	*R (%)	Found (mg L^−1^) Orange Juice	*R (%)
5	4.7	94	4.5	90	4.9	98
10	10.2	102	9.4	94	10	100
15	14.9	99	14.6	97	14.8	99

*R = Recovery.

**Table 4 sensors-20-05043-t004:** A comparison of analytical performance for optical determination of carrageenan.

Sensing Element	Transducer	Sensitivity (Δunits/decade)	R^2^	Dynamic Linear Range	Detection Limit (mg L^−1^)	Cost	Reference
MB	Reflectometry	377.5	0.980	80.0–5000.0 mg L^−1^	80.00	Low	Ling and Lee [31]
MB	Reflectometry	279.9	0.983	100.0–5000.0 mg L^−1^	100.00	Low	Ling and Lee [31]
MB	UV-Vis spectrophotometry			(0.2–2.0)×10^−3^%		High	Soedjak [24]
MB and Toluidine Blue	UV-Vis spectrophotometry		>0.996	2.0–60.0 mg L^−1^		High	Ziolkowska et al. [32]
MB	Fluorometry	−312.0	0.992	1.0–20.0 mg L^−1^	0.04	Low	Present study

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
