# Peer review of "Highly Sensitive Fluorescence Sensor for Carrageenan from a Composite Methylcellulose/Polyacrylate Membrane"

_sensors, 2020, doi:10.3390/s20185043_

Round 1
Reviewer 1 Report
I have reviewed the manuscript (sensors-849302) entitled “Highly Sensitive Fluorescence Sensor for Carrageenan from a Composite Methylcellulose/Polyacrylate Membrane” submitted to Sensors. The authors fabricated methylcellulose/ polyacrylate for membrane for carrageenan detection. The fabricated membrane was characterized by SEM, SEM-EDS, FT-IR. The sensor membrane showed detectability for carrageenan with a LOD value of 0.4 mg/L. Also, recovery of the spiked carrageenan into commercially available fruit juices was successfully demonstrated. This manuscript may provide an example of a fluorescence-based membrane sensor for carrageenan. However, the manuscript contains concerns regarding sensing mechanism, characterization, and significance. Thus, I recommend this manuscript for publication only after major revisions below.
- The mechanism of carrageenan detection is not clear. Why reaction with carrageenan causes color change as shown in Figure 1.
- The comparison in Table 4 is not fair. The authors should check the LOD value for carrageenan determined by other types of fluorescent sensors reported previously.
- The amount of immobilized methylene blue (MB) should be quantified.
- During the fabrication process for the sensor membrane, the DMF solution of MB was added to the THF solution of polymers. The subsequent drying process was carried out at 25 °C. Did the authors check on the amount of DMF remaining in the membrane by TGA?
- The chemical structure of Mc/PnBA should be provided in the main manuscript.
- The quality of the figures are below the level of scientific publication. Thus, the authors should increase their quality.
Author Response
Reviewer 1
I have reviewed the manuscript (sensors-849302) entitled “Highly Sensitive Fluorescence Sensor for Carrageenan from a Composite Methylcellulose/Polyacrylate Membrane” submitted to Sensors. The authors fabricated methylcellulose/ polyacrylate for membrane for carrageenan detection. The fabricated membrane was characterized by SEM, SEM-EDS, FT-IR. The sensor membrane showed detectability for carrageenan with a LOD value of 0.4 mg/L. Also, recovery of the spiked carrageenan into commercially available fruit juices was successfully demonstrated. This manuscript may provide an example of a fluorescence-based membrane sensor for carrageenan. However, the manuscript contains concerns regarding sensing mechanism, characterization, and significance. Thus, I recommend this manuscript for publication only after major revisions below.
- The mechanism of carrageenan detection is not clear. Why reaction with carrageenan causes color change as shown in Figure 1.
Answers:
Methylene blue (MB) was used as a carrageenan sensing fluorogenic dye and immobilised in the composite membrane. Carrageenan was detected by the electrostatic interaction between the cationic site of immobilised MB, i.e. alkylamino (=NR2+) functional group and anionic site of carrageenan, i.e. the negatively charged sulphate (−SO4−) functional group. As a result of such interaction, the blue coloured membrane turned purple due to the formation of a metachromatic complex. This will lead to a change in the fluorescence intensity as well as the colour of the sensor membrane. The MB could be immobilised in the Mc/PnBA membrane likely via, electrostatic interactions, hydrogen bonding and van der Waals forces between membrane structure and MB moleules. Figure 1. illustrates the interactions between the immobilised MB and carrageenan, and the chemical structure of Mc/PnBA composite membrane. Correction has been done in page 2, line 86-93 and Figure 1 of the revised manuscript.
- The comparison in Table 4 is not fair. The authors should check the LOD value for carrageenan determined by other types of fluorescent sensors reported previously.
Answers:
To our best knowledge, so far there has not been anyone reported on the carrageenan detection using fluorometry approach. Thus, it is not possible to compare with similar fluorescent sensor reported. However, it is still fair to compare with other optical sensors based on reflectance or UV-Vis spectrophotometry (Table 4) because they also employed immobilised MB as a sensing material [31]. Clearly, the fluorescence sensor based on MB immobilised Mc/PnBA composite membrane demonstrated LOD performance towards carrageenan with a significantly lower detection limit of approximately 2000 times smaller. Hence, the fluorescence-based carrageenan sensor has achieved low LOD comparable to that of most established methods, such as immunoassay and electrophoresis for the analysis of carrageenan in foodstuffs. Correction has been done in page 12, line 346-350 of the revised manuscript.
- The amount of immobilized methylene blue (MB) should be quantified.
Answers:
The concentration-effect study on the immobilised MB was carried out by immobilising MB dye onto the composite membrane at different concentrations from 0.01-0.15 mM, and the fluorescence intensity of the sensor membrane was measured at 675 nm. Thus, based on the intensity, the amount of immobilised MB had been qunatified.
- During the fabrication process for the sensor membrane, the DMF solution of MB was added to the THF solution of polymers. The subsequent drying process was carried out at 25 °C. Did the authors check on the amount of DMF remaining in the membrane by TGA?
Answers:
We agreed that the presence of DMF in the membrane resulted from inadequate drying may cause poor membrane property and this should be examined by TGA. However, from the preparation procedure, the amount of DMF used per membrane is about 1 mL (below 0.01 wt% of the membrane). Since the boiling point of DMF is just slightly higher than water, this minute amount of DMF used in the membrane should be evaporated off even at room temperature over the membrane drying process. And examination by TGA may not be necessary. This is confirmed by the good reproducibility of the sensor response. If the DMF has influence over the membrane integrity, the sensor response will not be reproducible.
- The chemical structure of Mc/PnBA should be provided in the main manuscript.
Answers:
We have included the chemical structure of the membrane in Figure 1. It illustrates the chemical reaction between the immobilised MB and carrageenan, and the chemical structure of Mc/PnBA composite membrane. Correction has been done in Figure 1 of the revised manuscript.
- The quality of the figures are below the level of scientific publication. Thus, the authors should increase their quality.
CORRECTION: We have improved the quality for Figure 2, Figure 3 and Figure 4.

Reviewer 2 Report
This paper titled “Highly Sensitive Fluorescence Optical Sensor for Carrageenan from a Composite Methylcellulose/Polyacrylate Membrane” described the detection of carrageenans using new fluorescent probe. There is no question about the potential usefulness of this work, and this manuscript contains a sufficient amount of new chemical information for the publication of Sensors in its present form
Author Response
Reviewer 2
This paper titled “Highly Sensitive Fluorescence Optical Sensor for Carrageenan from a Composite Methylcellulose/Polyacrylate Membrane” described the detection of carrageenans using new fluorescent probe. There is no question about the potential usefulness of this work, and this manuscript contains a sufficient amount of new chemical information for the publication of Sensors in its present form
RESPONSE: Thank you.

Reviewer 3 Report
The authors present an interesting method for determining the fluorescence carrageenan method in various substances. According to my opinion, the work does not contain relevant information to assess the quality of the method. Below comments
- Parameters characterizing the R2 method and sensitivity are not defined in the text
- It would be interesting to compare the measuring costs of the proposed method with existing methods used
- Can the proposed sensors be used many times?
Work clearly needs to be completed.
Author Response
Reviewer 3
The authors present an interesting method for determining the fluorescence carrageenan method in various substances. According to my opinion, the work does not contain relevant information to assess the quality of the method. Below comments.
- Parameters characterizing the R2 method and sensitivity are not defined in the text.
Answers:
The R2 is the linear correlation coefficient between the fluorescent intensity of the sensor and the concentrations of the carrageenan measured (Table 1 and Table 2). A value of R2=0.99 means that 99% of the two set of data is correlated in a certain range of concentrations. Thus, the concentration is strongly correlated with the fluorescent intensity in a linear manner. The R2 value can be calculated by correlation statistical program via the data processing software Excel Spreadsheet from a plot of fluorescent intensity against the carrageenan concentrations. The sensitivity of the sensor as mentioned in Table 1 and Table 2 is defined by the response of the sensor, i.e. the fluorescent intensity per unit change in concentration. Thus, this is a value of the slope from the linear plot of intensity vs carrageenan concentration over a certain concentration range. Thus, the larger the magnitude of the slope (or sensitivity), the more sensitive is the sensor response (fluorescence) towards changes in concentration. Correction has been done in page 10, line 298-308 of the revised manuscript.
- It would be interesting to compare the measuring costs of the proposed method with existing methods used
Answers:
Most of the current methods for carrageenan analysis involving instrumental procedures, e.g. electrophoresis, chromatography methods and spectrophotometric methods. Therefore, equipment and chemical consumables contributed to the high cost for carrageenan analysis. Our estimation of the sensor cost is about 10 times lower than that for an average cost of analysis via instrumentation techniques. Correction has been done in page 12, line 350-354 of the revised manuscript.
- Can the proposed sensors be used many times?
Answers:
The optical carrageenan sensor reported here is a single-use sensor strip of low cost for rapid analysis of carrageenan. Because of the relatively low cost needed, it is not necessary to reuse the sensor as the regeneration of the sensor for further use may take more time than the analysis. Correction has been done in page 10, line 291-294 of the revised manuscript.

Round 2
Reviewer 1 Report
Scholarly presentation has been improved after the revision. However, the quality of the figures needs further improvements.
Figure 1: Consider the aspect ratio.
Figures 5, 6, and 7: Remove frame, put tick marks on both axes, and use deep black for axes color.
Author Response
Reviewer 1
Scholarly presentation has been improved after the revision. However, the quality of the figures needs further improvements.
- Figure 1: Consider the aspect ratio.
Correction: Correction has been done in Figure 1 of the revised manuscript.
- Figures 5, 6, and 7: Remove frame, put tick marks on both axes, and use deep black for axes color.
Correction: Correction has been done in Figure 5, 6 and 7 of the revised manuscript.
